# Quality of Life Prediction on Walking Scenes Using Deep Neural Networks and Performance Improvement Using Knowledge Distillation

Thanasit Rithanasophon [1], Kitsaphon Thitisiriwech [1], Pittipol Kantavat [1,*], Boonserm Kijsirikul [1], Yuji Iwahori [2], Shinji Fukui [3], Kazuki Nakamura [4] and Yoshitsugu Hayashi [5]

1   Department of Computer Engineering, Faculty of Engineering, Chulalongkorn University, Pathumwan, Bangkok 10330, Thailand; 6270111921@student.chula.ac.th (T.R.); ck.kitsaphon@gmail.com (K.T.); boonserm.k@chula.ac.th (B.K.)
2   Department of Computer Science, Chubu University, Kasugai 487-8501, Japan; iwahori@isc.chubu.ac.jp
3   Faculty of Education, Aichi University of Education, Kariya 448-8542, Japan; sfukui@auecc.aichi-edu.ac.jp
4   Department of Civil Engineering, Meijo University, Nagoya 468-8502, Japan; knaka@meijo-u.ac.jp
5   Center for Sustainable Development and Global Smart City, Chubu University, Kasugai 487-8501, Japan; y-hayashi@isc.chubu.ac.jp
*   Correspondence: pittipol.k@chula.ac.th

**Abstract:** The well-being of residents is a top priority for megacities, which is why urban design and sustainable development are crucial topics. Quality of Life (QoL) is used as an effective key performance index (KPI) to measure the efficiency of a city plan's quantity and quality factors. For city dwellers, QoL for pedestrians is also significant. The walkability concept evaluates and analyzes the QoL in a walking scene. However, the traditional questionnaire survey approach is costly, time-consuming, and limited in its evaluation area. To overcome these limitations, the paper proposes using artificial intelligence (AI) technology to evaluate walkability data collected through a questionnaire survey using virtual reality (VR) tools. The proposed method involves knowledge extraction using deep convolutional neural networks (DCNNs) for information extraction and deep learning (DL) models to infer QoL scores. Knowledge distillation (KD) is also applied to reduce the model size and improve real-time performance. The experiment results demonstrate that the proposed approach is practical and can be considered an alternative method for acquiring QoL.

**Keywords:** quality of life; walking scene; walkability; semantic segmentation; object detection; deep convolutional neural networks; knowledge distillation

## 1. Introduction

Urban design is crucial for sustainable development in megacities. Fast-growing megacities without good urban planning can cause a lot of significant problems to residents such as traffic congestion and transportation accidents [1–5]. For this reason, policymakers need to realize the importance of sustainable development in urban design and plan both resident-friendly and environment-friendly cities.

Policymakers can use Quality of Life (QoL), one of the key performance indexes (KPI), to evaluate their urban design performance [6]. The concept of QoL [7] is to assess well-being, happiness of citizens, and individual needs including social interactions as a part of the society based on sustainable development [8–10]. According to numerous studies, people use the Quality-of-Life framework as a policy performance measurement for urban public transportation of megacities, especially in emerging countries, for example, Bangkok. The failed urban planning in Bangkok leads to severe transportation congestion that is caused by the car-oriented policies [1,2,11]. Thus, many researchers use Bangkok as a case of study for QoL, such as Alonso [12], who predicts land-use public transportation policy based on the satisfaction of dwellers, and Hayashi et al. [13] and Banister [14] proposed

that people should apply a QoL approach by implementing less car-oriented policies and encouraging the use of public transportation more.

The QoL of pedestrians is also a significant factor in the well-being of people living in the cities. Many studies propose the walkability concept to analyze and evaluate the QoL in a walking scene. Vichiensan and Nakamura [15] compare walking needs in Bangkok and Nagoya. The study highlights the importance of considering cultural and environmental factors when designing urban spaces that promote walking and active transportation. Nakamura [16] studies the correlation between walkability and QoL outcomes in car-oriented Asian cities. Vichiensan et al. [17] conclude that walking is essential to provide equitable access and mobility in a city. Sou et al. [18] developed a framework for evaluating street space by considering human emotions and values, aiming to improve communication between designers and stakeholders. Nakamura [19,20] highlights the importance of individual functions in pedestrian spaces and the effectiveness of design elements and proposes using virtual reality (VR) evaluation to improve the design process and create more effective pedestrian spaces.

Traditional tools used to evaluate Quality of Life (QoL) typically involve questionnaire-based surveys, which can be time-consuming and costly. However, the results obtained from these surveys are often limited to specific areas and timeframes, thus restricting their generalizability. Furthermore, integrating traditional QoL evaluation methods into other IT systems can present challenges and complexities. Some studies promoted new approaches to QoL by applying artificial intelligence (AI) to solve these limitations. Kantavat et al. [21] implemented deep convolutional neural networks (DCNNs), semantic segmentation, and object detection for extracting factors in transportation mobility from an image. Thitisiriwech et al. [22] proposed the Bangkok Urbanscapes dataset, the first labeled urban scene dataset in Bangkok, and models that have an excellent performance on semantic segmentation processing. Thitisiriwech et al. [23] proposed an AI approach using DCNNs and a linear regression model to extract information from images and infer the QoL score. Iamtrakul et al. [24] investigated how the built environment affects the QoL related to transportation using a combination of GIS and DL in the Sukhumvit district, Bangkok.

This study proposes a QoL evaluation method using the machine learning approach. There are two steps in the QoL assessment process. First, we conduct the information extraction using image processing, consisting of semantic segmentation and object recognition. Second, we feed the extracted information into the trained model to perform the QoL inference. In addition, we also propose a knowledge distillation (KD) framework to shorten the QoL inference time. The experiment results show that our proposed system is effective and can be considered in QoL evaluation for the walking scene.

The proposed novel method for evaluating Quality of Life (QoL) offers several distinct advantages compared to traditional approaches. One key advantage is the significant reduction in time and cost requirements. Unlike the conventional method, which is limited to specific survey areas and timeframes, our approach can be externally applied to evaluate QoL in different regions and at different points in time. Moreover, the proposed method is easily integrable into other IT systems that necessitate QoL evaluation or simulations.

Unlike previous research primarily concentrating on driving scenarios mentioned in the introduction, our study focuses on assessing QoL in the walking mode. This emphasis allows us to tailor our methodology to understand pedestrians' experiences better by considering that pedestrians observe and interact with their environment. We leverage virtual reality (VR) tools in the QoL questionnaire process to create a more immersive and realistic assessment. This enables us to capture the panoramic surroundings and the active engagement of pedestrians.

## 2. Literature Review

Many studies of QoL for the walking scene suggest the factors affecting walkability. Vichiensan and Nakamura [15] compared walking needs in Bangkok and Nagoya, finding that comfort and pleasurability are higher-level needs in both cities. Safety was

considered a higher-level need in Bangkok due to poorer street conditions. The study suggested that street improvements were needed to encourage walking in Bangkok and recover lost street activities in Nagoya. The study's inclusion of informal activities in the walkability evaluation provided practical street design insights for growing Asian cities. Nakamura [16] explored the relationship between walkability and QoL outcomes, particularly in car-oriented Asian cities. The study defined indicators of walkability QoL and surveyed 500 inhabitants of Nagoya city (Japan) using a questionnaire to assess their neighborhoods. Results showed that neighborhood street quality, particularly pleasurability, significantly affected QoL outcomes through their interrelationship. The study suggested that street quality was vital in land-use transport planning to improve QoL outcomes. Vichiensan et al. [17] summarized that walking was an important and sustainable mode of transportation that can provide equitable access and mobility in a city. However, to be enjoyable, convenient, and affordable, the walkway and nearby street must be attractive, vibrant, secure, uninterrupted, and well-protected from road traffic.

Some studies apply modern technology, such as AI and VR, for QoL and walkability. Sou et al. [18] proposed an AI and human co-operative evaluation (AIHCE) framework that facilitated communication design between designers and stakeholders based on human emotions and values for evaluating street space. The study suggested that the proposed framework can contribute to fostering people's awareness of streets as public goods, reflecting the essential functions of public spaces and the residents' values and regional characteristics, improving the city's sustainability. Nakamura [19,20] addressed the issue of pedestrian spaces needing more focus on individual functions and the effectiveness of design elements. The researcher proposed using VR evaluation to analyze the impact of sidewalk boundary space design on pedestrian space. The survey results showed a close relationship between boundary space design and pedestrian needs. The study also investigated the relationship between street environments, walking perceptions, and behaviors and found that VR evaluation reflects the sensitivity of walking willingness to the satisfaction of hierarchical walking needs.

Some studies conduct deep learning neural networks (DNNs) to improve the limitation of traditional QoL evaluation methods. Thitisiriwech et al. [23] suggested that the traditional QoL evaluation is costly and time-consuming. Hence, researchers proposed an AI approach using deep convolutional neural networks (DCNNs) and linear regression to predict QoL scores from driving-scene images, using DeepLab-v3+ and YOLO-v3 for knowledge extraction. The results also revealed insights into what makes Bangkokers feel happy or uncomfortable, such as wider roads and walkway spaces correlating with more delight and security and heavy traffic congestion reducing drivers' happiness. Iamtrakul et al. [24] used GIS and DL to analyze the effects of the built environment on Quality of Life in Transportation (QoLT) in Sukhumvit district, Bangkok. They found that individuals perceive QoLT differently depending on the physical characteristics of their environment. The study highlights the importance of understanding QoLT for urban planning and transportation development to achieve sustainable futures.

## 3. Related Theory

Deep learning (DL) is one of the machine learning techniques that is widely used, as it has been applied in numerous research studies. DL is a subfield of neural networks (NNs) and consists of an input layer, several hidden layers, and an output layer.

Deep supervised learning is a technique that uses labeled data to train the neural network model. The model then predicts output and compares the output with the target label to generate a loss value that will be used to update the weights for the model. There are various of approaches for this technique such as convolutional neural networks (CNNs), deep neural networks (DNNs), and recurrent neural networks (RNN).

This research uses these three neural networks. Image recognition is one of the convolutional neural network approaches that is described in Section 3.1, a DNN approach

is described in Section 3.2, and an RNN approach is described in Section 3.3. In addition, this research uses a bidirectional structure to implement the RNN as described in Section 3.4.

To improve the performance, this research employs knowledge distillation (KD) to reduce complexity and the consumed time of the model, as described in Section 3.5.

### 3.1. Image Recognition

This approach consists of two processes, object detection and semantic segmentation. The YOLOv4 model is selected for object detection process and the DDRNet-23-slim model is selected for semantic segmentation process. Details of each process are shown below.

- Object detection (YOLOv4)

A general object detection model consists of two parts; a backbone part which is pre-trained on ImageNet and a head part which is used to predict classes and bounding boxes of objects. Modern object detection adds more layers between the backbone and head, which is called the neck. The neck is usually used to collect feature maps from different stages.

YOLOv4 [25] is selected in this process. The model consists of a CSPDarknet53 backbone, an additional spatial pyramid pooling module along with a PANet path-aggregation neck, and YOLOv3 (anchor-based) head. However, there are some methods modified for efficient training and detection—CmBN, SAM, and PAN.

The accuracy of the model is 43.5% AP (65.7% AP50) for the MS COCO dataset at a real-time speed of ~65 FPS on Tesla V100.

- Semantic segmentation (DDRNet-23-slim)

Semantic segmentation is to classify a class for each pixel in the image. The result is a percentage, so the whole result is 100. Dilated convolutions are the standard for semantic segmentation because the convolutions with large dilations can handle very large images and high resolutions. Most state-of-the-art models are established from the ImageNet pre-trained backbone with the dilated convolutions. Figure 1 is an example of a dilated kernel in each rate.

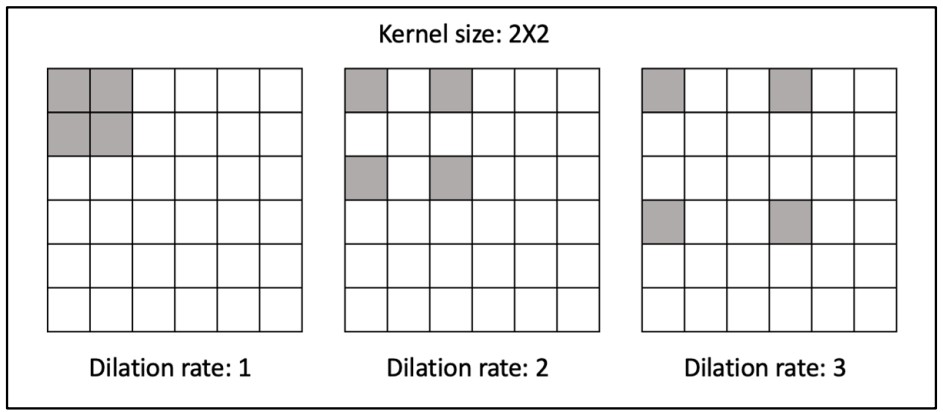

**Figure 1.** Example of dilated convolution kernel with rates 1, 2, and 3.

The DDRNet-23-Slim model [26] is selected in this process. The model has dual-resolution branches and multiple bilateral fusions for backbones.

Each branch has a bottleneck block at the latest layer and the low-resolution branch also has a deep aggregation pyramid pooling module before the pointwise concatenation layer that creates the result for the segmentation head to predict class labels.

This model is a smaller version of DDRNet-23 that is a trade-off between accuracy and processing speed (76.3 to 74.7 MIoU and 94 to 230 FPS).

### 3.2. Deep Neural Network

A deep neural network (DNN) consists of an input layer, hidden layers, and an output layer. At the hidden layer, there are stacks of several layers. Therefore, DNN is more complex than traditional neural networks and requires longer durations of time to train the model.

This research uses a Convolutional 1 Dimension (CONV1D) and fully connected neural network. The characteristics of the network are shown in Figures 2 and 3.

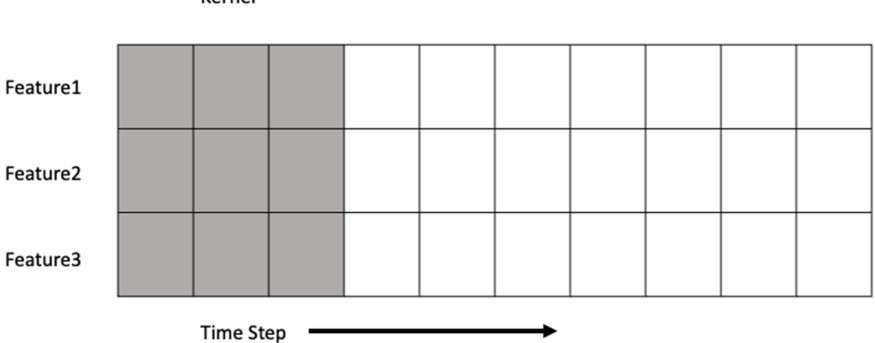

**Figure 2.** The characteristics of CONV1D neural network.

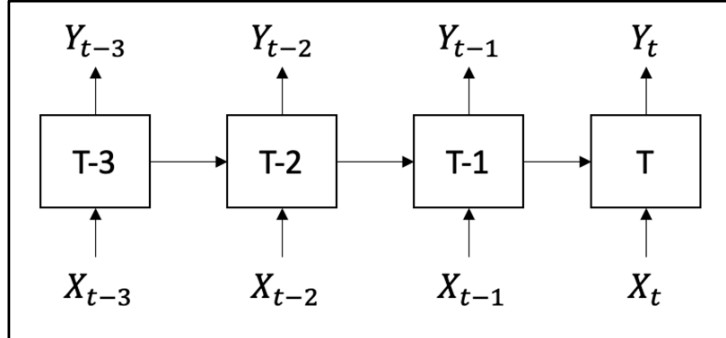

**Figure 3.** Recurrent Neural Network.

CONV1D is one of the convolutional neural networks whose kernel slides along only one dimension to generate a feature map. This CNN type is proper for time-series data where each feature needs to be processed at the same time through time steps.

A fully connected neural network is the neural network whose current node is connected to every node from the prior layer. At the process step, the node uses mathematic function to calculate the input to be output for the next layer. The function is called the "activation function" that is selected depending on an expected output type. Examples of traditional functions include Rectified Linear Unit (ReLU), Sigmoid, Hyperbolic tangent (tanh), and SoftMax.

### 3.3. Recurrent Neural Network

A recurrent neural network (RNN) is one of the traditional neural networks. This network is usually trained on sequential data, such as text and video. The output from the previous step will be used as the input for the next step.

This research uses two RNN models, Long Short-Term Memory units (LSTM) and Gated Recurrent Unit (GRU), as part of the model. These two models can deal with the vanishing gradient problem encountered by traditional RNNs [27].

LSTM and GRU both have the same main component, gates as shown in Equation (1). There are four types of gates: (1) update gate, (2) relevance gate, (3) forget gate, and (4) output gate. Each of these gates has different functions as follows:

- Update gate ($\Gamma_u$)—To weigh the importance of previous information to cell state.
- Relevance gate ($\Gamma_r$)—To decide if the cell should ignore previous information or not.
- Forget gate ($\Gamma_f$)—To identify if cell state should be reset.
- Output gate ($\Gamma_o$)—To determine how much information the cell should carry.

$$\Gamma = \sigma(Wx_t + Ua_{t-1} + b) \tag{1}$$

LSTM contains all four types of gates but GRU contains only two types of gates, the update gate and relevance gate. Figure 4a,b show the architecture of LSTM and GRU, respectively.

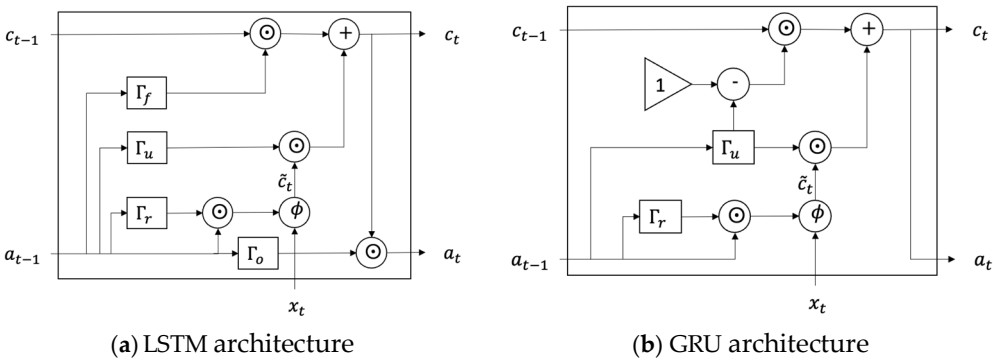

(**a**) LSTM architecture                (**b**) GRU architecture

**Figure 4.** LSTM architecture (**a**) and GRU architecture (**b**) [27].

### 3.4. Bidirectional Network

A bidirectional network is a network architecture that incorporates information from both the forward and backward layers of the model, where the outputs of these layers are typically concatenated at each time step. This structure is commonly employed with RNNs. This research uses bidirectional LSTM and bidirectional GRU networks to enhance the prediction performance. The bidirectional nature of these models proves beneficial in specific tasks, such as language translation, where the two-way processing aids in capturing contextual information effectively.

### 3.5. Knowledge Distillation

Knowledge distillation (KD) [28] is to distill or move knowledge from a complex pre-trained model into a smaller model without significant reduction in performance. An overview of KD is shown in Figure 5.

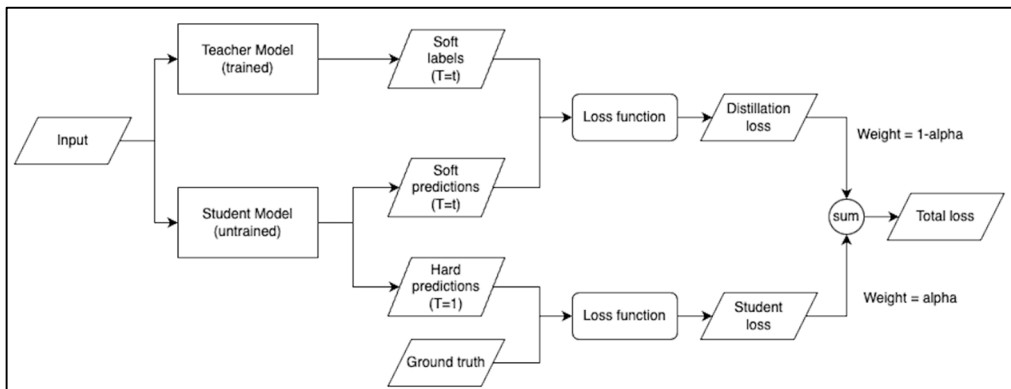

**Figure 5.** An overview of knowledge distillation.

Firstly, to obtain soft labels, the previous pre-trained model (teacher model) is used to make predictions, then the results are divided by temperature (T). Secondly, soft predictions are calculated by predictions from a trained smaller model (student model) and the results

are divided by temperature (T). When comparing soft labels and soft predictions, this creates a loss result which is called distillation loss. Student loss, another loss result, is from comparing predictions of trained student models to the ground truth. To obtain the total loss of the proposed system in this paper, we multiply these two results of loss with weights and then sum them. The weight of distillation loss is one minus alpha. The weight of student loss is alpha. A pseudo code of this process is shown in Figure 6.

```
Constant value is temperature and alpha

Function distillationLoss (Argument teacher output, Argument student output) {
        Divide argument teacher output by temperature to get teacher soft label
        Divide argument student output by temperature to get student soft label

        Return Compare teacher soft label with student soft label by distillation loss function
}

Function calculateTotalLoss (Argument student loss result, Argument distillation loss result) {
        Return alpha x argument student loss result + (1 - alpha) x argument distillation loss result
}

Function knowledgeDistillation (Argument input, Argument target) {
        Predict student output by student model with argument input
        Compare student output with argument target by student loss function to get student loss result

        Predict teacher output by pre-trained teacher model with argument input
        Send predicted student output and predicted teacher output to the distillationLoss function to get the distillation loss result

        Send student loss result and distillation loss result to the calculateTotalLoss function to get total loss result
        Use total loss result to adjust weights of student model
}
```

**Figure 6.** A pseudo code of knowledge distillation.

In summary, our proposed method consists of two steps: QoL inference and performance improvement. The QoL inference step involves the application of image recognition, DNN, RNN, and bidirectional network approaches to generate QoL inference. In the performance improvement step, knowledge distillation (KD) is applied to reduce complexity and increase the model's performance, resulting in reduced processing speed. However, it is essential to note that there is a trade-off in this process, as the accuracy may be slightly decreased.

## 4. Proposed Methods

This section provides details of the processes in our proposed QoL evaluation methods. Section 4.1 describes the overview of our framework, consisting of two steps, QoL inference and KD processes. Section 4.2 describes the dataset used in this research and shows the label used to compare with the prediction results. Section 4.3 describes the feature extraction from images using image recognition models. Section 4.4 describes the architecture used to train the QoL score model and shows the hyperparameters used to configure the architecture. Section 4.5 shows how to process the input and output data before the training step. Section 4.6 describes the training strategy and how to group the dataset. Section 4.7 shows the measuring equation used to evaluate the results. Section 4.8 shows the configuration, hyperparameters, and small architecture used as the student model and final architecture that is used as the teacher model for the KD process.

### 4.1. Our Framework

Our framework is composed of two processes; the first one is the QoL inference process, and the second one is the KD process.

The QoL inference process is described in Sections 4.3–4.7; the YOLOv4 model and the DDR-Net-23-Slim model extract the image information from the video. Then, we train a DL model using the extracted information as an input label and the data from the questionnaire survey as an output label. Figure 7 shows the overview of the QoL inference process.

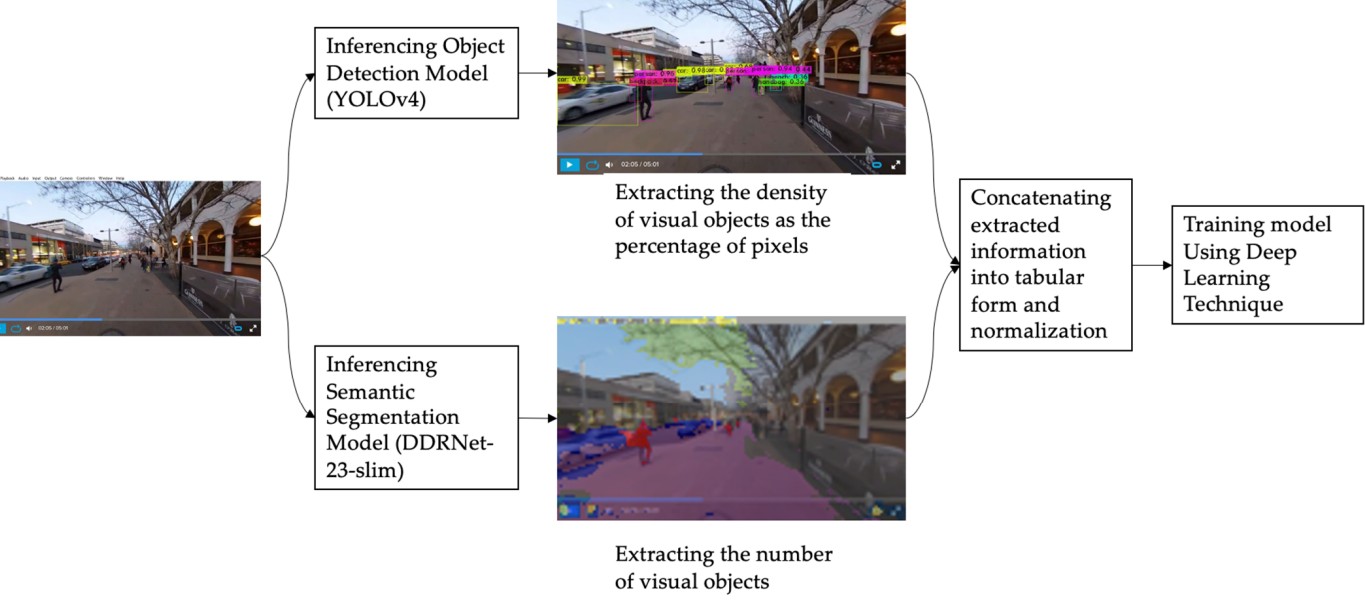

**Figure 7.** The overview of "the" QoL inference step.

The KD process is described in Section 4.8. We provide the configuration for this step, including the model architecture used as the student model. Figure 8 shows the overview of the KD process.

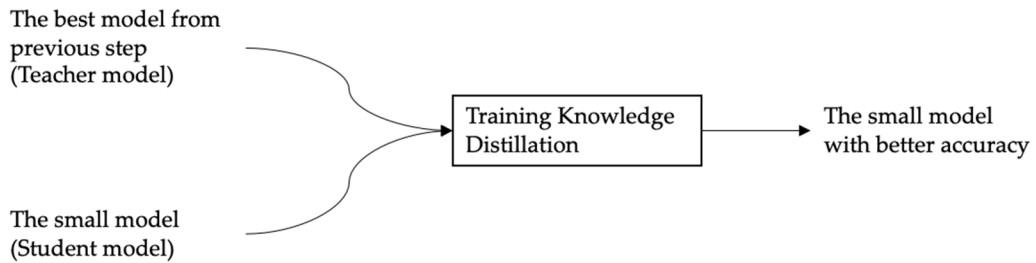

**Figure 8.** The overview of "the" knowledge distillation step.

### 4.2. The QoL Dataset

As an input, we use recorded videos of the walking scene from four cities, Canberra, Bangkok, Brisbane, and Sakae. Each city consists of three one-minute scenes. Therefore, the total number of scenes in the input is 12 scenes. The camera used in data collection is RICOH Theta S, recorded at 24 fps for Bangkok, Brisbane, and Canberra and recorded at 30 fps for Sakae.

Table 1 presents the evaluation of walkability indicators, including factors such as Building, Facility, Roof, Bench, Stall, Walking or Sitting People, Footpath Width, Pedestrian, Brightness, Bicycle Parking, Roadside Tree, Electric Pole, Parking, Zebra Crossing, and Traffic Volume. These factors, rated on a scale of 1–10, were derived from the considerations of street design and walking needs, as outlined in the reference paper [19].

**Table 1.** The 15 factors of evaluated score for each respondent in the questionnaire.

| | |
|---|---|
| Building | Building |
| | Facility |
| | Roof |
| Activity | Bench |
| | Stall |
| | Walking or Sitting People |
| Footpath | Footpath Width |
| | Pedestrian |
| | Brightness |
| Installation | Bicycle Parking |
| | Roadside Tree |
| | Electric Pole |
| Roadway | Parking |
| | Zebra Crossing |
| | Traffic Volume |

The questionnaire survey involved displaying virtual scenes from each location to the interviewees using a VR Google tool. The survey occurred at Nakamura Lab, Department of Civil Engineering, Meijo University. There are several benefits to using virtual reality technology for conducting the questionnaire survey. Firstly, it allows interviewees to immerse themselves in the surrounding environment of each scene, enabling them to perceive the scene better and understand the QoL in those settings compared to simply watching video clips on a computer monitor. Secondly, interviewees can control their viewing direction, enabling them to focus on specific elements such as the walking path, sky, trees, or buildings according to their preferences. Finally, because the scenes were collected from different countries, it would be impractical to take interviewees to the actual locations physically. Utilizing VR tools offers a cost-effective solution for surveying such circumstances.

*4.3. Feature Extraction Using Image Recognition*

This subsection details a feature extraction approach to extract features from prior-recorded videos. This approach consists of two processes, object detection and semantic segmentation.

Each video is captured into pictures frame by frame and then passed to an object detection model, YOLOv4, and a semantic segmentation model, DDRNet-23-Slim. The results of the YOLOv4 model are shown in Figure 9, which visualizes the object detected with bounding boxes, and DDRNet-23-Slim in Figure 10, which visualizes highlight colors at each pixel.

The YOLOv4 model and the DDR-Net-23-Slim model extract the image information from the video. The extracting results from YOLOv4 consist of 17 classes (as shown in Table 2), while the extracting results from DDR-Net-23-Slim consist of 19 classes (as shown in Table 3).

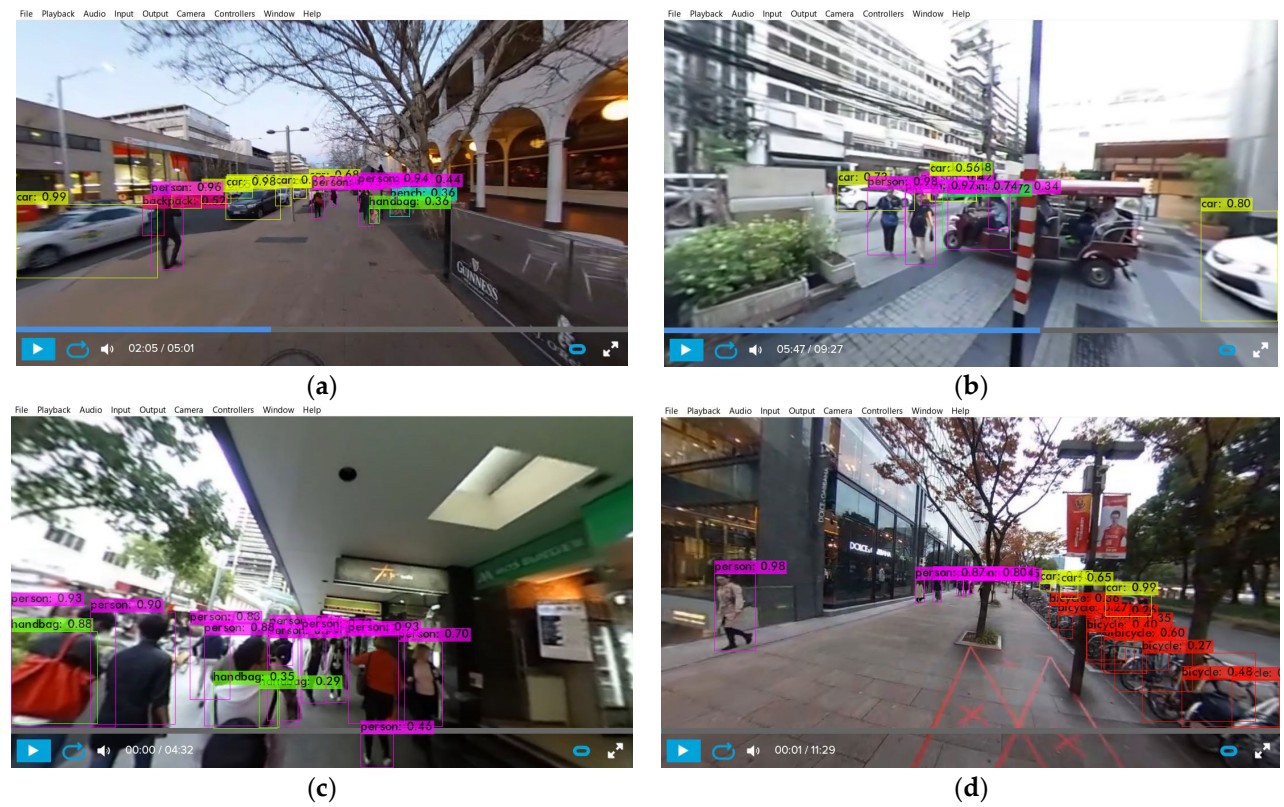

**Figure 9.** Example of the results from YOLOv4. (**a**) Canberra; (**b**) Bangkok; (**c**) Brisbane; (**d**) Sakae.

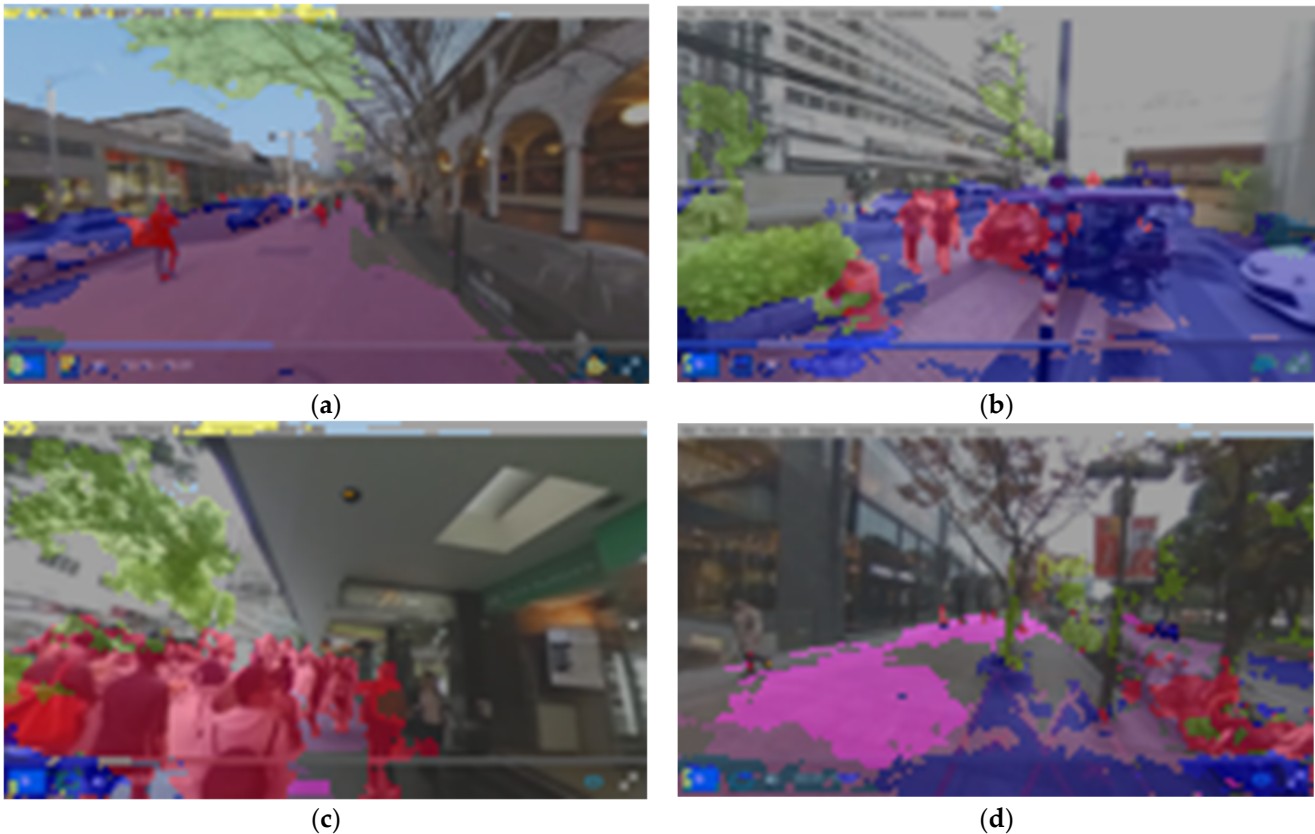

**Figure 10.** Example of the results from DDRNet-23-Slim. (**a**) Canberra; (**b**) Bangkok; (**c**) Brisbane; (**d**) Sakae.

**Table 2.** The result classes from YOLOv4 model.

| | |
|---|---|
| 1. Person | 10. Traffic light |
| 2. Bicycle | 11. Fire hydrant |
| 3. Car | 12. Stop sign |
| 4. Motorbike | 13. Parking meter |
| 5. Aeroplane | 14. Bench |
| 6. Bus | 15. Backpack |
| 7. Train | 16. Umbrella |
| 8. Truck | 17. Handbag |
| 9. Boat | |

**Table 3.** The result classes from DDRNet-23-Slim model.

| Colors | | | | | | |
|---|---|---|---|---|---|---|
| Class | 1. Road | 2. Sidewalk | 3. Building | 4. Wall | 5. Fence | 6. Pole | 7. Traffic light |
| Colors | | | | | | |
| Class | 8. Traffic sign | 9. Vegetation | 10. Terrain | 11. Sky | 12. Person | | 13. Rider |
| Colors | | | | | | |
| Class | 14. Car | 15. Truck | 16. Bus | 17. Train | 18. Motorcycle | | 19. Bicycle |

### 4.4. The QoL Prediction Model Architecture

We use the TensorFlow framework to implement DNN for QoL prediction models. Figure 11 shows the QoL prediction model architecture overview, consisting of 4 boxes. The top box illustrates the architecture's first layer, in which two selectable options are CONV1D or fully connected with the tanh activation function and SoftMax. The second box presents the recurrent network layer, in which four selectable options are LSTM, bidirectional LSTM (bi LSTM), GRU, and bidirectional GRU (bi GRU). Then, the dropout layer is adopted, as shown in the third box. Finally, the stack of five consequences, fully connected with bias and dropout layers, is implemented.

For the top box's first option, CONV1D, the kernel slides along 1 dimension only to generate a feature map; this part uses 13 filter kernels with the size of $64 \times 64$. Another option is fully connected with the tanh activation function and SoftMax; the tanh activation function is represented as Equation (2), and a SoftMax layer is represented as Equation (3).

$$Tanh = \frac{(e^x - e^{-x})}{(e^x + e^{-x})} \tag{2}$$

$$SoftMax = \frac{e^{z_i}}{\sum_{j=1}^{K} e^{z_j}} \tag{3}$$

where

| | |
|---|---|
| $z_i$ | = input at current position; |
| $e^{z_i}$ | = standard exponential function for an input at current position; |
| $K$ | = number of total input vector; |
| $e^{z_j}$ | = standard exponential function for an input at position $j$. |

Four optional RNNs can be selected for the second box. The last box has a dashed box with five numbers; the highest number must include all the lower numbers. For instance, box number 3 comprises boxes 1 to 3, which have a fully connected layer, dropout, and another fully connected layer.

Based on our hyperparameter tuning, we have determined that the first two layers in this structure should consist of 128 units each. Additionally, we have applied

a dropout layer to mitigate model overfitting and enhance generalization performance. Figures 12 and 13 are examples of QoL prediction model architecture with hyperparameters in each layer and trainable parameters of the models, including the dropout layer (20% or 0.2 dropouts for all models).

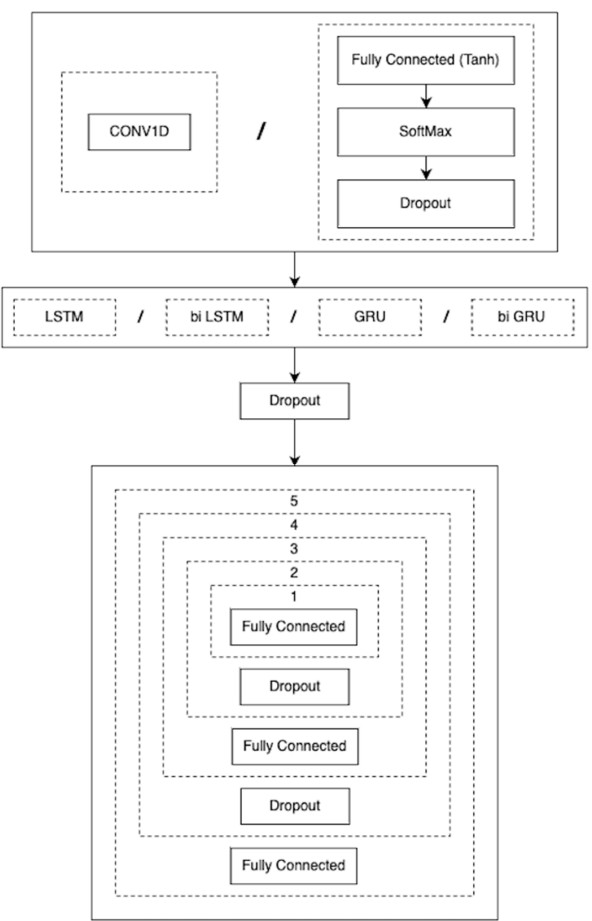

**Figure 11.** Overview of QoL prediction model architecture.

```
Layer (type)               Output Shape              Param #
=================================================================
input (InputLayer)         [(None, 15, 37)]          0

conv1d (Conv1D)            (None, 13, 64)            7168

lstm (LSTM)                (None, 128)               98816

dropout (Dropout)          (None, 128)               0

dense (Dense)              (None, 128)               16512

dropout_1 (Dropout)        (None, 128)               0

dense_1 (Dense)            (None, 128)               16512

dropout_2 (Dropout)        (None, 128)               0

dense_2 (Dense)            (None, 128)               16512

dense_3 (Dense)            (None, 15)                1935

=================================================================
Total params: 157,455
Trainable params: 157,455
Non-trainable params: 0
```

**Figure 12.** The architecture of the QoL prediction model selecting the options of CONV1D for the first box and LSTM for the second box.

```
Layer (type)                 Output Shape              Param #
=================================================================
input (InputLayer)           [(None, 15, 37)]          0

dense (Dense)                (None, 15, 128)           4736

tf.nn.softmax (TFOpLambda)   (None, 15, 128)           0

dropout (Dropout)            (None, 15, 128)           0

lstm (LSTM)                  (None, 128)               131584

dropout_1 (Dropout)          (None, 128)               0

dense_1 (Dense)              (None, 128)               16512

dropout_2 (Dropout)          (None, 128)               0

dense_2 (Dense)              (None, 128)               16512

dropout_3 (Dropout)          (None, 128)               0

dense_3 (Dense)              (None, 128)               16512

dense_4 (Dense)              (None, 15)                1935

=================================================================
Total params: 187,791
Trainable params: 187,791
Non-trainable params: 0
```

**Figure 13.** The QoL prediction model architecture selecting the options of fully connected with the tanh activation function and SoftMax for the first box and LSTM for the second box.

### 4.5. Data Preprocessing

We utilized the dataset of one-minute scenes, dividing each scene into 30 frames. In total, there are 360 frames, with 4 cities and 3 scenes per city. To reduce the input size for the QoL prediction model, we calculated the average of features extracted from consecutive video frames. These averaged features were then grouped into 30 samples per scene, serving as input features for the QoL prediction model. The model's output labels were derived from survey results obtained from 50 respondents. The input features for the QoL prediction model are illustrated in Figure 14, consisting of three vectors. The first vector represents the output of the YOLOv4 model, containing 17 features as shown in Table 2. The second vector represents the output of the DDRNet-23-Slim model, comprising 19 features as shown in Table 3. These features were normalized to match the range of values from the output of YOLOv4. Additionally, we appended unique pseudo-demographic numbers to identify the interviewees' answers. Figure 15 provides an overview of the QoL prediction model, demonstrating its overall architecture and flow.

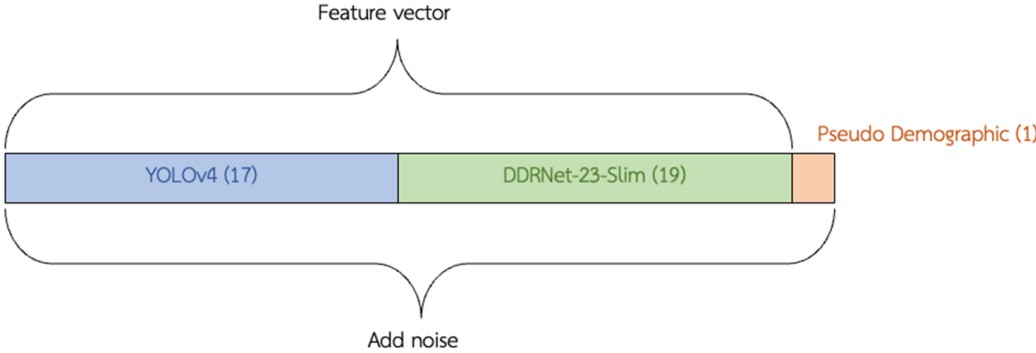

**Figure 14.** The input features for the QoL prediction model.

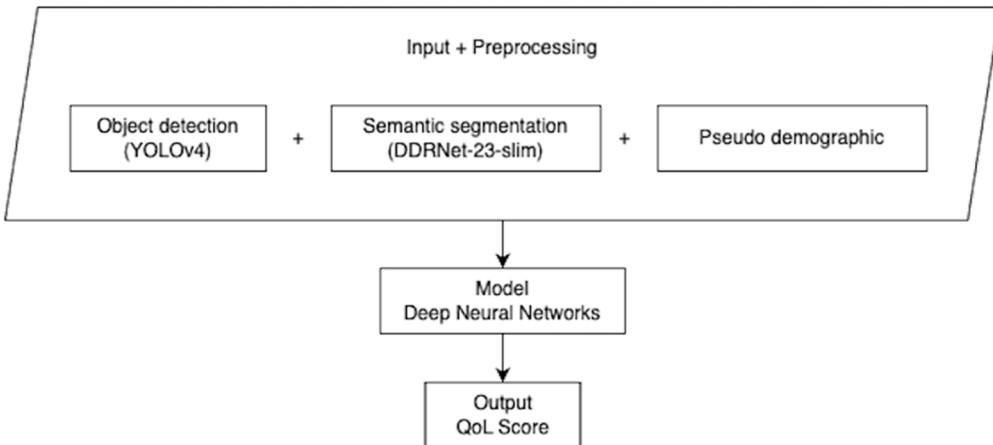

**Figure 15.** Input features and output labels in the training process of the QoL prediction model.

*4.6. Experiment Sets and Dataset Splitting*

We set up two experiment groups to evaluate our framework and models: the within-the-city and across-the-cities groups. For the within-the-city group, we arrange the training and testing data from the same city, while for the across-the-cities group, we arrange the data by shuffling along the different cities, as shown in Tables 4 and 5, respectively.

**Table 4.** Training and testing data for the within-the-city experiment group.

| Experiment Set | Training Data | Testing Data |
|:---:|:---:|:---:|
| 1 | Canberra Scene2, Canberra Scene3 | Canberra Scene1 |
| 2 | Canberra Scene1, Canberra Scene3 | Canberra Scene2 |
| 3 | Canberra Scene1, Canberra Scene2 | Canberra Scene3 |
| 4 | Bangkok Scene2, Bangkok Scene3 | Bangkok Scene1 |
| 5 | Bangkok Scene1, Bangkok Scene3 | Bangkok Scene2 |
| 6 | Bangkok Scene1, Bangkok Scene2 | Bangkok Scene3 |
| 7 | Brisbane Scene2, Brisbane Scene3 | Brisbane Scene1 |
| 8 | Brisbane Scene1, Brisbane Scene3 | Brisbane Scene2 |
| 9 | Brisbane Scene1, Brisbane Scene2 | Brisbane Scene3 |
| 10 | Sakae Scene2, Sakae Scene3 | Sakae Scene1 |
| 11 | Sakae Scene1, Sakae Scene3 | Sakae Scene2 |
| 12 | Sakae Scene1, Sakae Scene2 | Sakae Scene3 |

**Table 5.** Training and testing data for the across-the-cities experiment group.

| Experiment Set | Training Data | Testing Data |
|:---:|:---:|:---:|
| 1 | Bangkok Scene all, Brisbane Scene all, Sakae Scene all | Canberra Scene all |
| 2 | Canberra Scene all, Brisbane Scene all, Sakae Scene all | Bangkok Scene all |
| 3 | Canberra Scene all, Bangkok Scene all, Sakae Scene all | Brisbane Scene all |
| 4 | Canberra Scene all, Bangkok Scene all, Brisbane Scene all | Sakae Scene all |

### 4.7. Model Evaluation

Mean Square Error (MSE) is a statistical method for measuring the error between prediction output and ground truth information. The MSE equation is shown in Equation (4), where $n$ denotes the number of labels.

$$MSE = \frac{1}{n}\sum_{i=1}^{n}(Y_i - \hat{Y}_i)^2 \tag{4}$$

### 4.8. The Knowledge Distillation Architecture

The KD consists of the teacher model, a model with high complexity and accuracy, and the student model, a small structure; the teacher model helps the student model train easily. Moreover, the model trained by this technique will have lower accuracy than the teacher model but higher accuracy than the training itself. Also, the size of the model is smaller than the teacher model. Therefore, KD is applied to reduce the model size and improve the real-time performance of the teacher model. The structure is shown in Figure 16, and the configuration is shown in Figure 17. The student model structure used for training is shown in Figure 18, and the configuration is shown in Figure 19. In our implementation, we have set the temperature parameter to 10 and the alpha parameter to 0.1, the original values recommended in the reference source [29]. Additionally, we have used MSE as the loss function for both the student and distillation models.

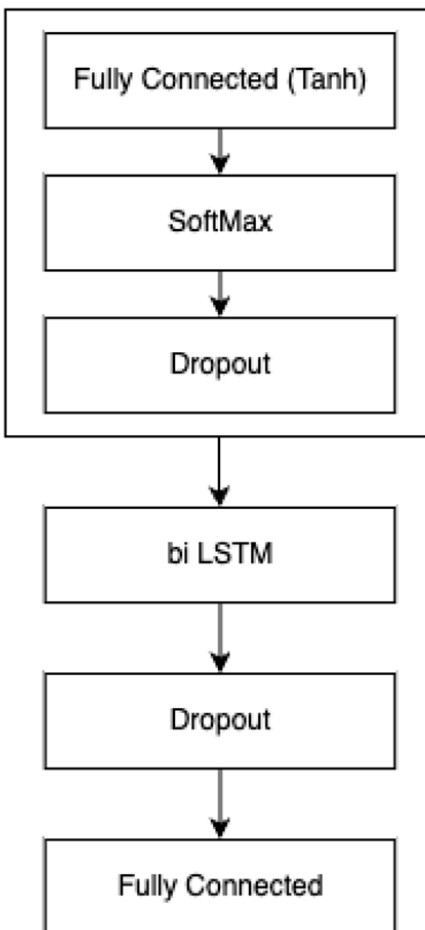

**Figure 16.** An architecture of the teacher model training in the knowledge distillation process.

```
Layer (type)                Output Shape          Param #
=================================================================
 input (InputLayer)         [(None, 15, 37)]        0

 dense (Dense)              (None, 15, 128)         4736

 tf.nn.softmax (TFOpLambda) (None, 15, 128)         0

 dropout (Dropout)          (None, 15, 128)         0

 bidirectional (Bidirectiona (None, 256)            263168
 l)

 dropout_1 (Dropout)        (None, 256)             0

 dense_1 (Dense)            (None, 128)             32896

 dense_2 (Dense)            (None, 15)              1935

=================================================================
Total params: 302,735
Trainable params: 302,735
Non-trainable params: 0
```

**Figure 17.** A configuration of the teacher model training in the knowledge distillation process.

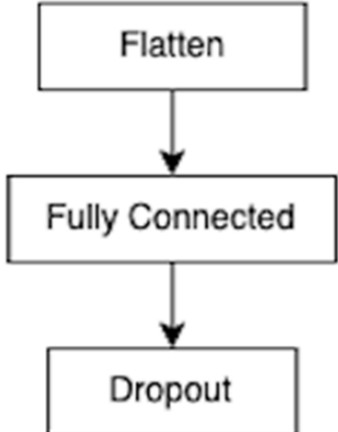

**Figure 18.** An architecture of the student model training in the knowledge distillation process.

```
Layer (type)                Output Shape          Param #
=================================================================
 input (InputLayer)         [(None, 15, 37)]        0

 flatten (Flatten)          (None, 555)             0

 dense (Dense)              (None, 256)             142336

 dropout (Dropout)          (None, 256)             0

 dense_1 (Dense)            (None, 15)              3855

=================================================================
Total params: 146,191
Trainable params: 146,191
Non-trainable params: 0
```

**Figure 19.** A configuration of the student model training in the knowledge distillation process.

## 5. Experiment Results

To verify the effectiveness of our proposed method for QoL inference and knowledge distillation (KD) processes, we conducted experiments using Google Collaboratory. Google Collaboratory is a cloud computing service offering a Python environment and access to high-performance CPU and GPU resources. Specifically, our experiments utilized an Intel(R) Xeon(R) CPU @ 2.20 GHz as the CPU and a Tesla T4 GPU [30].

The results of our experiments are presented in this section, which consists of two subsections. Section 5.1 showcases the outcomes of the QoL prediction model, while Section 5.2 focuses on the results obtained from applying the KD process to the models.

### 5.1. The Result of the QoL Prediction Model

We set the experiments at 10,000 epochs for training with a 0.001 learning rate. Figures 20 and 21 show the training loss of the within-the-city experiment group and across-the-cities experiment group, respectively. The figures show MSE changes plotted as a graph every 50 epochs, with a total of 200 plotted points.

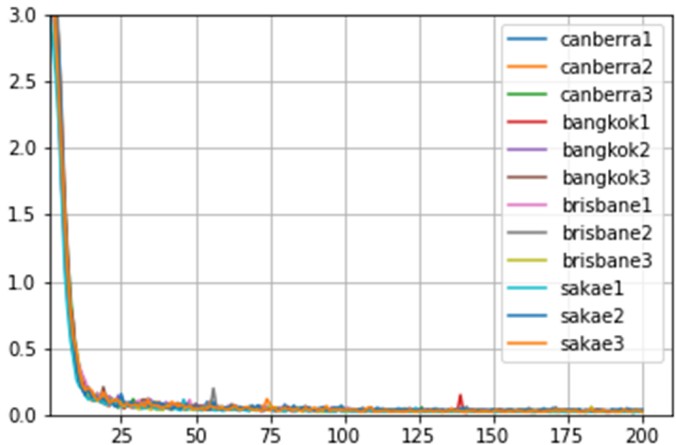

**Figure 20.** An example of loss graphs showing changes in each epoch of model training for the within-the-city experiment group.

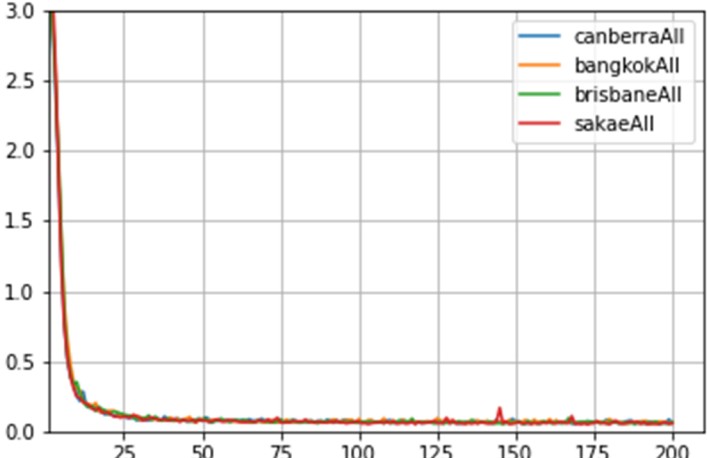

**Figure 21.** An example of loss graphs showing changes in each epoch of model training for the across-the-cities experiment group.

The test results with the lowest MSE from the within-the-city experiment group and the across-the-cities experiment group setups are shown in Tables 6 and 7, respectively. The results in Table 6 indicate that the fully connected tanh activation function combined with bidirectional LSTM is the best model, winning 8 out of 12 within-the-city experiment

sets. The results in Table 7 also indicate that the fully connected tanh activation function combined with bidirectional LSTM is the best model, winning in the across-the-cities experiment sets.

**Table 6.** The test results of the within-the-city experiment group.

| Training Dataset | Testing Dataset | Model Name | MSE |
|---|---|---|---|
| Canberra Scene2, Canberra Scene3 | Canberra Scene1 | Tanh bi LSTM 1 | $7.41 \times 10^{-3}$ |
| Canberra Scene1, Canberra Scene3 | Canberra Scene2 | Tanh bi LSTM 1 | $8.35 \times 10^{-3}$ |
| Canberra Scene1, Canberra Scene2 | Canberra Scene3 | Tanh bi LSTM 1 | $7.92 \times 10^{-3}$ |
| Bangkok Scene2, Bangkok Scene3 | Bangkok Scene1 | CONV1D bi LSTM 1 | $8.11 \times 10^{-3}$ |
| Bangkok Scene1, Bangkok Scene3 | Bangkok Scene2 | Tanh bi LSTM 1 | $7.55 \times 10^{-3}$ |
| Bangkok Scene1, Bangkok Scene2 | Bangkok Scene3 | CONV1D bi LSTM 1 | $8.62 \times 10^{-3}$ |
| Brisbane Scene2, Brisbane Scene3 | Brisbane Scene1 | Tanh bi LSTM 1 | $7.19 \times 10^{-3}$ |
| Brisbane Scene1, Brisbane Scene3 | Brisbane Scene2 | Tanh bi GRU 1 | $9.27 \times 10^{-3}$ |
| Brisbane Scene1, Brisbane Scene2 | Brisbane Scene3 | Tanh bi LSTM 1 | $7.55 \times 10^{-3}$ |
| Sakae Scene2, Sakae Scene3 | Sakae Scene1 | Tanh bi GRU 1 | $6.80 \times 10^{-3}$ |
| Sakae Scene1, Sakae Scene3 | Sakae Scene2 | Tanh bi LSTM 1 | $1.14 \times 10^{-2}$ |
| Sakae Scene1, Sakae Scene2 | Sakae Scene3 | Tanh bi LSTM 1 | $8.90 \times 10^{-3}$ |

**Table 7.** The test results of the across-the-cities experiment group.

| Training Dataset | Testing Dataset | Model Name | MSE |
|---|---|---|---|
| Bangkok Scene all, Brisbane Scene all, Sakae Scene all | Canberra Scene all | Tanh bi LSTM 1 | $9.73 \times 10^{-3}$ |
| Canberra Scene all, Brisbane Scene all, Sakae Scene all | Bangkok Scene all | Tanh bi LSTM 1 | $1.20 \times 10^{-2}$ |
| Canberra Scene all, Bangkok Scene all, Sakae Scene all | Brisbane Scene all | Tanh bi LSTM 1 | $1.16 \times 10^{-2}$ |
| Canberra Scene all, Bangkok Scene all, Brisbane Scene all | Sakae Scene all | Tanh bi LSTM 1 | $1.17 \times 10^{-2}$ |

*5.2. The Results of the Knowledge Distillation*

We apply the KD process to the fully connected tanh activation function combined with bidirectional LSTM as it is the model with the lowest MSE in both the within-the-city and the across-the-cities experiment groups. The distillation results of the within-the-city group are shown in Tables 8 and 9, while the distillation results of the across-the-cities group are shown in Tables 10 and 11.

**Table 8.** The MSE results of the within-the-city group.

| Training Dataset | Testing Dataset | Student Model MSE | Teacher Model MSE | Distilled Model MSE |
|---|---|---|---|---|
| Canberra Scene2, Canberra Scene3 | Canberra Scene1 | 2.92 | $7.16 \times 10^{-3}$ | $6.16 \times 10^{-2}$ |
| Canberra Scene1, Canberra Scene3 | Canberra Scene2 | 3.50 | $9.08 \times 10^{-3}$ | $5.24 \times 10^{-2}$ |
| Canberra Scene1, Canberra Scene2 | Canberra Scene3 | 3.62 | $8.26 \times 10^{-3}$ | $2.77 \times 10^{-2}$ |
| Bangkok Scene2, Bangkok Scene3 | Bangkok Scene1 | 3.40 | $6.09 \times 10^{-3}$ | $3.30 \times 10^{-2}$ |
| Bangkok Scene1, Bangkok Scene3 | Bangkok Scene2 | 3.49 | $8.46 \times 10^{-3}$ | $6.86 \times 10^{-2}$ |
| Bangkok Scene1, Bangkok Scene2 | Bangkok Scene3 | 4.04 | $6.93 \times 10^{-3}$ | $8.91 \times 10^{-2}$ |
| Brisbane Scene2, Brisbane Scene3 | Brisbane Scene1 | 3.93 | $7.74 \times 10^{-3}$ | $5.07 \times 10^{-2}$ |
| Brisbane Scene1, Brisbane Scene3 | Brisbane Scene2 | 3.666 | $7.26 \times 10^{-3}$ | $5.99 \times 10^{-2}$ |
| Brisbane Scene1, Brisbane Scene2 | Brisbane Scene3 | 3.567 | $6.53 \times 10^{-3}$ | $5.34 \times 10^{-2}$ |
| Sakae Scene2, Sakae Scene3 | Sakae Scene1 | 2.968 | $8.73 \times 10^{-3}$ | $4.80 \times 10^{-2}$ |
| Sakae Scene1, Sakae Scene3 | Sakae Scene2 | 3.632 | $9.45 \times 10^{-3}$ | $5.08 \times 10^{-2}$ |
| Sakae Scene1, Sakae Scene2 | Sakae Scene3 | 3.641 | $5.14 \times 10^{-3}$ | $2.93 \times 10^{-2}$ |
| Average MSE | | 3.530 | $7.57 \times 10^{-3}$ | $5.20 \times 10^{-2}$ |

**Table 9.** The computational time results of the within-the-city group.

| Training Dataset | Testing Dataset | Teacher Model Computational Time (ms) | Distilled Model Computational Time (ms) |
|---|---|---|---|
| Canberra Scene2, Canberra Scene3 | Canberra Scene1 | 7.518 | 1.271 |
| Canberra Scene1, Canberra Scene3 | Canberra Scene2 | 7.076 | 0.861 |
| Canberra Scene1, Canberra Scene2 | Canberra Scene3 | 6.804 | 0.893 |
| Bangkok Scene2, Bangkok Scene3 | Bangkok Scene1 | 6.830 | 0.908 |
| Bangkok Scene1, Bangkok Scene3 | Bangkok Scene2 | 7.100 | 0.868 |
| Bangkok Scene1, Bangkok Scene2 | Bangkok Scene3 | 7.023 | 0.836 |
| Brisbane Scene2, Brisbane Scene3 | Brisbane Scene1 | 7.051 | 1.306 |
| Brisbane Scene1, Brisbane Scene3 | Brisbane Scene2 | 6.867 | 0.893 |
| Brisbane Scene1, Brisbane Scene2 | Brisbane Scene3 | 6.991 | 0.948 |
| Sakae Scene2, Sakae Scene3 | Sakae Scene1 | 10.182 | 1.295 |
| Sakae Scene1, Sakae Scene3 | Sakae Scene2 | 7.208 | 0.882 |
| Sakae Scene1, Sakae Scene2 | Sakae Scene3 | 8.152 | 1.203 |
| Average Computation Time | | 7.400 | 1.014 |

**Table 10.** The MSE results of the across-the-cities group.

| Training Dataset | Testing Dataset | Student Model MSE | Teacher Model MSE | Distilled Model MSE |
|---|---|---|---|---|
| Bangkok Scene all, Brisbane Scene all, Sakae Scene all | Canberra Scene all | 3.81 | $9.65 \times 10^{-3}$ | 1.23 |
| Canberra Scene all, Brisbane Scene all, Sakae Scene all | Bangkok Scene all | 3.55 | $1.13 \times 10^{-2}$ | 1.28 |
| Canberra Scene all, Bangkok Scene all, Sakae Scene all | Brisbane Scene all | 3.51 | $1.14 \times 10^{-2}$ | 1.14 |
| Canberra Scene all, Bangkok Scene all, Brisbane Scene all | Sakae Scene all | 3.87 | $1.04 \times 10^{-2}$ | 1.26 |
| Average MSE | | 3.69 | $1.07 \times 10^{-2}$ | 1.23 |

**Table 11.** The computational time results of the across-the-cities group.

| Training Dataset | Testing Dataset | Teacher Model Computational Time (ms) | Distilled Model Computational Time (ms) |
|---|---|---|---|
| Bangkok Scene all, Brisbane Scene all, Sakae Scene all | Canberra Scene all | 2.330 | 0.299 |
| Canberra Scene all, Brisbane Scene all, Sakae Scene all | Bangkok Scene all | 2.308 | 0.291 |
| Canberra Scene all, Bangkok Scene all, Sakae Scene all | Brisbane Scene all | 2.392 | 0.292 |
| Canberra Scene all, Bangkok Scene all, Brisbane Scene all | Sakae Scene all | 2.302 | 0.322 |
| Average Computation Time | | 2.333 | 0.301 |

Table 8 shows the results of MSE for each experiment set in the within-the-city group. The results reveal that the distilled models have a lower MSE than the student model but a higher MSE than the teacher model. The conclusion shows that the KD MSE results are as expected for the within-the-city group.

The prediction results obtained from the teacher and distilled models are presented in Figure 22. We have selected examples for each factor from the evaluated QoL score factors. The training set consists of Canberra Scene1 and Canberra Scene2, while the testing set consists of Canberra Scene3. The visualization of the results indicates that both models' expected and predicted results are consistent. Additionally, it is observed that the computation time and model size are smaller for the distilled model compared to the teacher model in the within-the-city experiment groups.

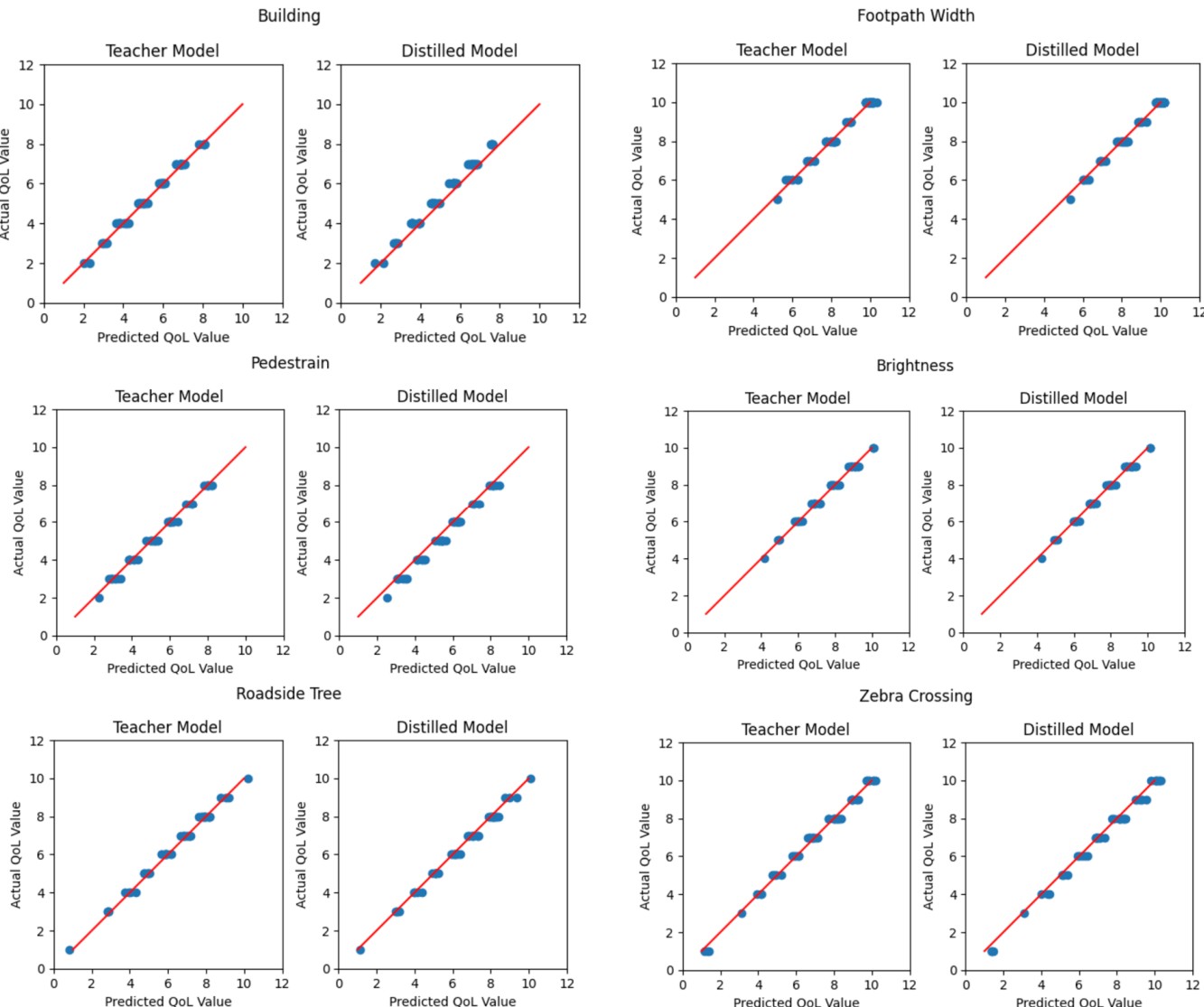

**Figure 22.** The comparison graph plots between the teacher and distilled models' actual results (*Y*-axis) and predicted results (*X*-axis) selected from a testing set of Canberra Scene3, within-the-city experiment groups.

Table 9 shows the results of the computation time for each experiment set in the within-the-city group. The comparison of the results shows that the distilled model computes the output while consuming a lower computational time than the teacher model. The conclusion indicates that the KD computational time results are as expected for the within-the-city group.

Table 10 shows the results of MSE for each experiment set in the across-the-cities groups. The results show that the distilled models have a lower MSE than the student model but a higher MSE than the teacher model. The conclusion is that the KD MSE results are as expected for the across-the-cities group.

Figure 23 illustrates the prediction outcomes of both the teacher and distilled models. To evaluate the QoL score factor for the across-the-cities experiment group, we chose Bangkok, Brisbane, and Sakae Scene all as the training set, while the Canberra Scene all served as the testing set. The visualization reveals that the teacher model's expected and predicted results closely align with the diagonal line. However, the results from the distilled model are scattered more than the within-the-city experiment group. Moreover, the distilled model demonstrates reduced computation time and size compared to the teacher model.

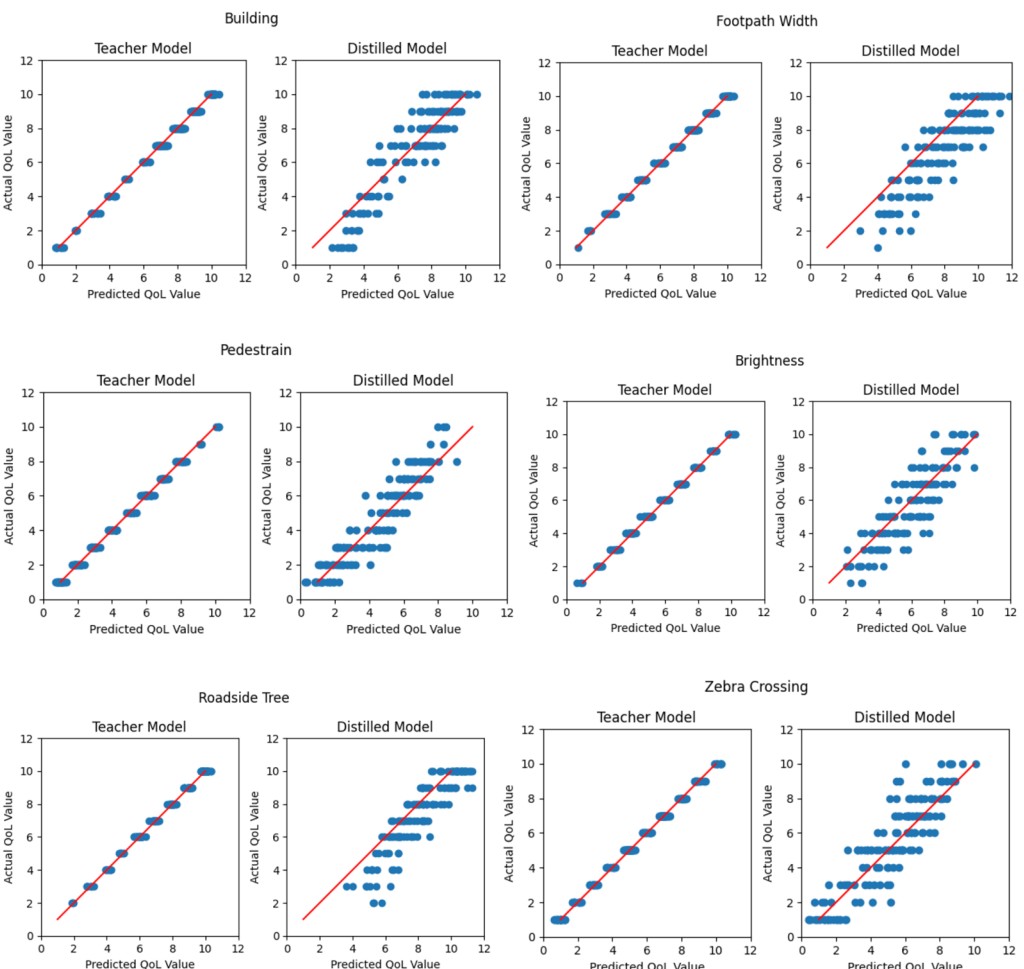

**Figure 23.** The comparison graph plots between the teacher and distilled models' actual results (*Y*-axis) and predicted results (*X*-axis) selected from a testing set of Canberra Scene all, across-the-cities experiment groups.

Table 11 indicates the results of the computation time for each experiment set in the across-the-cities groups. The results show that the distilled model outperforms the teacher model due to lower computational time. The KD computational time results are as expected for the across-the-cities group.

## 6. Discussion

### 6.1. The Result of the QoL Prediction Model

The fully connected tanh activation function combined with bidirectional LSTM is the best model, providing the lowest MSE in almost all experiment sets. The lowest MSEs are $7.19 \times 10^{-3}$ with Brisbane Scenes1 and 2 as the training dataset and Scene3 as the testing dataset for the within-the-city experiment group as shown in Table 6, and $9.73 \times 10^{-3}$ with Bangkok Scene all, Brisbane Scene all, and Sakae Scene all as the training dataset and Canberra Scene all as the testing dataset for the across-the-cities experiment group as shown in Table 7.

### 6.2. The Results of the Knowledge Distillation

The knowledge distillation (KD) is applied to the best model architecture from the previous part (fully connected tanh activation function combined with bidirectional LSTM) to reduce complexity and the time consumed for QoL prediction by transferring knowledge to the student model. The comparison results show that the distilled model is less time-consuming than the complex model but outputs the QoL with a higher MSE. Additionally,

the distilled model boasts a smaller size, approximately half that of the teacher model, as depicted in Figures 17 and 19.

The MSE from the within-the-city experiment group (as shown in Table 8) shows that the distilled model has an average MSE of $5.20 \times 10^{-2}$. The MSE from the across-the-cities group (as shown in Table 10) shows that the distilled model has an average MSE of 1.23. Furthermore, the results of the computational time (as shown in Tables 9 and 11, respectively) show that the distilled model can output the QoL value at a lower average computation time, reducing from 7.40 to 1.01 for the within-the-city experiment group and from 2.33 to 0.30 for the across-the-cities group.

## 7. Conclusions

Megacities prioritize the well-being of their residents, making urban design and sustainable development crucial. Quality of Life (QoL) is an effective key performance index (KPI) used to measure a city plan's efficiency. Walkability is significant for city dwellers, and the traditional questionnaire survey approach to evaluate it is costly, time-consuming, and limited in its evaluation area. To address these limitations, we propose using artificial intelligence (AI) technology to evaluate walkability data collected through a questionnaire survey using virtual reality (VR) tools.

The proposed method involves using deep convolutional neural networks (DCNNs) for knowledge extraction and deep learning (DL) models to infer QoL scores. In our case, the experiment results indicate that a smaller model outperforms a larger model due to a low number of layers for both experiments that train the data within and across the city. KD can aid the model by using a large and complicated model to train a smaller model, resulting in higher accuracy and a lower time consumption. In summary, the experiment results show that the proposed approach is practical and can be an alternative method for acquiring QoL scores.

**Author Contributions:** Conceptualization, T.R., K.T., P.K., B.K. and Y.I.; methodology, T.R., K.T. and P.K.; software, T.R. and K.T.; validation, T.R., K.T., P.K. and B.K.; formal analysis, T.R., K.T. and P.K; investigation, T.R., K.T., P.K., B.K., Y.I. and S.F.; resources, Y.I., S.F. and K.N..; data curation, Y.I., S.F. and K.N.; writing—original draft preparation, T.R.; writing—review and editing, T.R., P.K. and B.K.; visualization, T.R.; supervision, P.K., B.K., Y.I. and S.F.; project administration, T.R., K.T., P.K. and Y.H.; funding acquisition, Y.I. and Y.H. All authors have read and agreed to the published version of the manuscript.

**Funding:** This research was funded by Science and Technology Research Partnership for Sustainable Development (SATREPS), Japan Science and Technology Agency (JST)/Japan International Cooperation Agency (JICA) "Smart Transport Strategy for Thailand 4.0" (Chair: Yoshitsugu Hayashi, Chubu University, Japan) under Grant JPMJSA1704, and by Japan Society for the Promotion of Science (JSPS) Grant-in-Aid for Scientific Research (C)(20K11873) and by Chubu University Grant.

**Institutional Review Board Statement:** Not applicable.

**Informed Consent Statement:** Not applicable.

**Data Availability Statement:** Not applicable.

**Conflicts of Interest:** The authors declare no conflict of interest.

## Abbreviations

The following abbreviations are used in this manuscript:

| | |
|---|---|
| QoL | Quality of Life |
| QoLT | Quality of Life Transportation |
| KPI | Key Performance Index |
| AI | Artificial Intelligence |
| DCNNs | Deep Convolutional Neural Networks |
| DL | Deep Learning |
| NN | Neural Network |

| RNN | Recurrent Neural Network |
| CNN | Convolutional Neural Network |
| DNN | Deep Neural Network |
| KD | Knowledge Distillation |
| YOLO | You Only Look Once |
| DDRNet | Deep Dual-resolution Networks |
| CSPNet | Cross Stage Partial Network |
| SPP | Spatial Pyramid Pooling |
| PANet | Path Aggregation Network |
| CmBN | Cross mini-Batch Normalization |
| SAT | Self-adversarial training |
| SAM | Spatial Attention Module |
| mIoU | mean Intersection over Union |
| FPS | Frames Per Second |
| CONV1D | Convolutional 1 Dimension |
| ReLU | Rectified Linear Unit |
| Tanh | Hyperbolic Tangent |
| LSTM | Long Short-Term Memory units |
| GRU | Gated Recurrent Unit |
| MSE | Mean Square Error |

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
