# Peer review of "Quality of Life Prediction on Walking Scenes Using Deep Neural Networks and Performance Improvement Using Knowledge Distillation"

_electronics, doi:10.3390/electronics12132907_

Round 1
Reviewer 1 Report
The manuscript presents a very interesting proposal to evaluate walkability of cities (4) using data collected through a questionnaire survey using Virtual Reality tools and a set of AI algorithms related to object detection and improving the computational time consumed
In general, the manuscript is well written and the objectives of the work are clear. The organization of the paper need some improvements, and some parts should be better explained. Some considerations are noted below:
At the end part of the methodology section, a general description of the manuscript content is useful for readers, maybe a paragraph containing this information can be added.
The contribution of the paper should be clearly stated in the introduction section
Section 2 and Section 3 have the same title, I think they are different section, may one is a subsection of the other. Please, verify this error.
Virtual reality is mentioned in the abstract and some parts of the manuscript, but it is not explained and it is easy to understand how it was used.
Authors need to verify the use of acronyms, some time they are used and others the full definition is used.
Were Figures 2, 3, 5 created by the authors? Or do they belong to another work? Due to some authoring issues it is better to only insert own figures.
There is some basic information that can be reduced, especially in Section 3. Note that the length of the paper is high and it can be reduced dropping basic concept, and more specifically some figures.
For the object detection model, why Yolo V4 was selected? Others algorithms were tested? Please, justify why this model was considered in the proposed solution. The same question about the Semantic segmentation model…
Give more details about the dataset used in the test. There are 12 videos of one-minute length but what are the content categories of these videos. How the scenes were chosen? IS this total information (12 minutes) enough to determine a model (considering training and testing phases)?
Table present some criteria set. How it was determined? Is there any reference for it ?
A more detailed information about Data processing subsection (4.3). Fig. 19 and 20 are not deeply explained. Each scene is 2-seconds length?
Table 9 shows the computational time before and after the destilled model is used. Any information about the reduction of the model size?
Model Evaluation- Section. Is Mean Square Error the sole metric being used to evaluate a complex solution? I think, this is main concern about the proposed solution. More performance metrics should be considered in the general evaluation of the proposed solution, and there are some of them in the literature.
The results presented look good, but I think they could be expanded using more performance evaluation metrics, to evaluate each of the steps of the proposed method.
Author Response
Thank you for your comments. Please see the provided responses as below.
At the end part of the methodology section, a general description of the manuscript content is useful for readers, maybe a paragraph containing this information can be added.
>> We added the summary of section 3 at line 261-267 to help readers understand the methodology better.
The contribution of the paper should be clearly stated in the introduction section
>> In the revised version, we have highlighted the contribution of our method in the last paragraph of the introduction section. This paragraph explicitly emphasizes the unique value and impact our approach brings to the field of study.
Section 2 and Section 3 have the same title, I think they are different section, may one is a subsection of the other. Please, verify this error.
>> Apologies for the confusion. In the revised version, we have made the necessary correction and changed the title of Section 2 to "Literature Review" instead of the previous title. Thank you for bringing it to our attention.
Virtual reality is mentioned in the abstract and some parts of the manuscript, but it is not explained and it is easy to understand how it was used.
>> To provide a more comprehensive understanding of the virtual reality technology utilized in our research, we have expanded the details in the last paragraph of section 4.2, titled "The QoL Dataset." This paragraph now delves into the specifics of the virtual reality technology employed and elaborates on its advantages and benefits in the context of our study.
Authors need to verify the use of acronyms, some time they are used and others the full definition is used.
>> We have already revised the paper by adjusting the acronyms in the same pattern as your recommended. Thank you.
Were Figures 2, 3, 5 created by the authors? Or do they belong to another work? Due to some authoring issues it is better to only insert own figures.
>> Our team created all figures presented in this paper. However, we removed specific figures during the revision process to reduce the paper's overall length. Please refer to the revised version of the article to view the remaining figures that have been included.
There is some basic information that can be reduced, especially in Section 3. Note that the length of the paper is high and it can be reduced dropping basic concept, and more specifically some figures.
>> After careful consideration, we have reduced the in-depth details in Subsection 3.1. You will find a more concise and summarized presentation of the relevant information in the revised version. This approach maintains the essential points while ensuring the overall readability and length of the paper are optimized. Please refer to the revised version to review the updated summary details in Subsection 3.1.
For the object detection model, why Yolo V4 was selected? Others algorithms were tested? Please, justify why this model was considered in the proposed solution. The same question about the Semantic segmentation model…
>> Among the available object detection techniques, there are other models such as R-CNN or R-CNN. However, after comparing these models, we opted to use the YOLO model due to its faster processing speed and effective accuracy. We have prior experience with the YOLO family and have used it in other research projects. While there are many versions of YOLO available, for our experiment conducted in 2022, we found YOLO version 4 suitable for our task. In terms of semantic segmentation or real-time tasks, considering the stable models at the time of our experiment, we selected DDRNet-23-slim due to its fast computation speed and high frame rate (fps).
Ref:
-https://paperswithcode.com/sota/object-detection-on-pascal-voc-2007
-https://paperswithcode.com/sota/real-time-semantic-segmentation-on-cityscapes
Give more details about the dataset used in the test. There are 12 videos of one-minute length but what are the content categories of these videos. How the scenes were chosen? IS this total information (12 minutes) enough to determine a model (considering training and testing phases)?
>> About the categories of the videos, since our study focuses on the quality of life (QoL) in walking scenes for urban planning purposes, all the video scenes fall under the category of walking in urban city scenes. As this research is a part of the SATREP project (more details about SATREPS in the Funding section), we selected cities based on the residence of our research members. The chosen cities include Bangkok, Sakae, Canberra, and Brisbane. Based on the MSE (Mean Squared Error) obtained from the experimental results, we believe the information provided is sufficient for model determination.
Table present some criteria set. How it was determined? Is there any reference for it ?
>> We are not sure that you mentioned about the factors presented in Table 1, "15 factors of evaluated score for each respondent in the questionnaire"? If yes, they were derived from walkability indicators, which encompass street design and walking needs. These indicators were determined based on the relevant literature cited in reference paper number 19. “歩道境界空間デザインを考慮した VR歩行空間評価”
To provide clarity for the reader, we have included detailed information in Section 4.2, specifically in the second paragraph, lines 302-306, explaining the origins of the walkability factors.
A more detailed information about Data processing subsection (4.5). Fig. 19 and 20 are not deeply explained. Each scene is 2-seconds length?
>> We provided the information of the vectors that are used to be an input of the QoL inference model at line 377-389.
Table 9 shows the computational time before and after the distilled model is used. Any information about the reduction of the model size?
>> We added Figure 17 to compare the size with Figure 19. In addition, we added more information in Subsection 6.2.
Model Evaluation- Section. Is Mean Square Error the sole metric being used to evaluate a complex solution? I think, this is main concern about the proposed solution. More performance metrics should be considered in the general evaluation of the proposed solution, and there are some of them in the literature.
>> We decided to employ the Mean Squared Error (MSE) metric to assess the accuracy and deviation of the predicted value. While MSE effectively captures these aspects, we acknowledge the importance of further enhancing the evaluation of our experimental results. To address this, we have incorporated an additional method in the discussion section, specifically lines 474-483 and lines 500-509, which includes a plot depicting the actual value versus the predicted value. This visual representation provides a more comprehensive assessment and validation of our findings.
The results presented look good, but I think they could be expanded using more performance evaluation metrics, to evaluate each of the steps of the proposed method.
>> The same explanation to comment #13
Reviewer 2 Report
The study's authors created a model to evaluate the impact of low-carbon land use transport development on the Quality of Life (QOL) index, using Bangkok as an example. They verified the model's accuracy with data from Bangkok, Thailand. Below are some recommendations for the authors to consider:
1. Clarify the formula used to define QOL in this study.
2. To highlight any discrepancies, display a graph that compares the actual and predicted QOL values (x-axis for actual values and y-axis for predicted values).
3. Explain the algorithm utilized and how to determine the optimal parameters.
4. Although the authors compared the proposed method's results with other methods, they still need to address the reasons for any inaccuracies in the predictions.
Moderate editing of the English language is required.
Author Response
Thank you for your comments. Please see the provided responses as below.
1. Clarify the formula used to define QOL in this study.
>> Compared to the traditional articles mentioned in the introduction section, our approach presents a distinct methodology for predicting the Quality of Life (QoL) value. Firstly, instead of relying on the authors' subjective identification of factors influencing QoL, we employ image recognition techniques, such as semantic segmentation and object detection, to extract factors directly from video images. This approach eliminates the need for predefined factors and allows interviewees to freely focus on relevant aspects of the images without being guided by specific questionnaire questions.
Secondly, while traditional articles calculate factor weights based on survey responses to derive QoL equations, our approach utilizes Deep Learning techniques to construct a QoL prediction model. This involves training the input video images directly against interviewees' answers. Consequently, our method eliminates implicit factors and formulas, offering a black-box model that encapsulates the formulas within its framework.
In summary, our proposed method diverges from traditional approaches by avoiding explicit factors and formulas. Instead, it presents a block-box model that employs Deep Learning techniques to establish QoL predictions using video images and interviewees' responses.
2. To highlight any discrepancies, display a graph that compares the actual and predicted QOL values (x-axis for actual values and y-axis for predicted values).
>> As per your suggestion, we have included Figure 22, titled "The comparison graph plots between the teacher and distilled models' actual results (Y-axis) and predicted results (X-axis) selected from a testing set of Canberra Scene 3, Within-the-city experiment groups," and Figure 23, titled "The comparison graph plots between the teacher and distilled models' actual results (Y-axis) and predicted results (X-axis) selected from a testing set of Canberra Scene all, Across-the-cities experiment groups," along with their corresponding explanations in the specified sections. We greatly appreciate your valuable recommendation to enhance the comprehensiveness of our article.
3. Explain the algorithm utilized and how to determine the optimal parameters.
>> As detailed in Section 4.1, "Our Framework," our methodology involves several key steps:
- A collection of video scenes is required, and it is essential to note that a more significant number and diversity of videos contribute to the robustness of the trained model.
- A questionnaire survey related to the specific aspects of QoL of interest is necessary. Again, more surveys with diverse backgrounds and socioeconomic factors result in improved QoL prediction outcomes.
- We extract information from the videos using image recognition techniques and construct the QoL prediction model by training it on the QoL values obtained from the questionnaire survey.
The overall process is summarized in Figure 7, which provides an overview of the QoL inference step.
While we have achieved the QoL prediction model with optimal accuracy, there may be instances where a high-speed model is required for real-time tasks. In such cases, we employ the Knowledge Distillation (KD) process. This involves using the acquired model as the Teacher model and constructing a Student model for practical usage. Figure 8 summarizes the knowledge distillation step, offering an overview.
Regarding the determination of optimal parameters in each step, the selected and tuned parameters are obtained through the machine learning and distillation processes elucidated in the article. It is important to note that while observing the final weight of each parameter within the model is possible, explaining it is challenging due to the inherent nature of Machine Learning.
4. Although the authors compared the proposed method's results with other methods, they still need to address the reasons for any inaccuracies in the predictions.
>> As outlined in Section 4.4, "The QoL Prediction Model Architecture," we have provided comprehensive details regarding the available options of Machine Learning (ML) models used for QoL prediction. Given the nature of ML model training, it is essential to note that there is no definitive guarantee that any particular model will outperform others across all datasets. However, it is possible to identify a model that exhibits superior performance compared to alternative choices through a consensus-based approach. Determining the best ML model involves analyzing empirical results, and it can be challenging to make a definitive statement. The assessment of model superiority relies on factors such as the model's suitability to the problem's complexity, the input data's diversity, and the overall compatibility of the model's components. Therefore, while providing a clear-cut verdict on the best model is complex, the selection process is driven by empirical evidence and considerations related to problem complexity, data diversity, and the effective functioning of individual components within the model.
The primary factor contributing to inaccuracies in the predictions stems from the diverse backgrounds of the interviewees when answering the QoL values. As each interviewee brings their unique perspective and experiences, the QoL values assigned to the same scene are expected to vary significantly. Consequently, the predicted QoL value serves as a collective summary derived from the inputs of all participants. Since individual perspectives influence QoL values, it is vital to recognize the inherent subjectivity in assessing and predicting QoL. While efforts are made to capture a comprehensive understanding of QoL through the participation of multiple interviewees, variations in their responses inevitably contribute to the challenge of achieving precise predictions. Therefore, it is crucial to acknowledge that the predicted QoL value represents a synthesis of diverse perspectives rather than a definitive and universally applicable measure.
Reviewer 3 Report
Please see the attachment!!

Author Response
Thank you for your comments. Please see the provided responses as below.
-
This study proposes a QoL evaluation method using the Machine Learning approach. However, few works of literature or description were mentioned in this paper about the motivation for using proposed AI method. Please strengthen the motivation and describe why proposed AI methods are better than the traditional tools or other AI approachs.
>> We have already written more details about our motivation in the 4th paragraph of the introduction of the revised version.We also highlight why our proposed method is better than the traditional tools and other AI tools in the two last paragraphs of the introduction section.
-
Please describe the main contribution of this article.
>> In the revised version, we have highlighted the contribution of our method in the two last paragraphs of the introduction section. This paragraph explicitly emphasizes the unique value and impact our approach brings to the field of study. -
Line128 has same title as line 81, please clarify.
>> Apologies for the confusion. In the revised version, we have made the necessary correction and changed the title of Section 2 to "Literature Review" instead of the previous title. Thank you for bringing it to our attention. -
The article only mentions the GPU, it is recommended that fully describe the hardware used in this research.
>> We have already provided more details about the hardware used in our research in Section 5, “Experiment Results.” -
Please describe the composition of the dataset. Please describe how many images in total and how to allocate the training data set and test data set?
>> We have provided additional details about the dataset in Subsection 4.5, "Data Preprocessing." This section now offers a comprehensive overview of the dataset used in our study. Additionally, in Subsection 4.6, "Experiment Sets and Dataset Splitting," we have included specific information about the training and test datasets and the process of splitting the dataset for experimentation purposes. These additions aim to offer readers a thorough understanding of the dataset utilized in our study and how it was prepared and divided for training and testing. -
If possible, it is recommended to add some case studies so that readers can have a deeper understanding.
>> Our proposed approach, which utilizes AI for assessing Quality of Life (QoL) in walkability, is considered pioneering, and limited case studies are available for the reader to reference. However, to better understand traditional QoL evaluation in the context of walkability, we recommend reviewing the papers [15-20], which delve into the subject in greater detail. Additionally, we suggest referring to the papers [21-24] to explore past developments in AI for QoL assessment. These resources will give the reader valuable insights into traditional and AI-based approaches for evaluating QoL in walkability.
Round 2
Reviewer 1 Report
The manuscript was modified according to the reviewer comments. Please, author should give a second review of the paper format. I think, it can be accepted in its current presentation.
Reviewer 2 Report
The authors have made the requested revisions to the paper, based on the suggestions provided. Therefore, I recommend accepting the article in its current form.